# A Microfluidic-Based Investigation of Bacterial Attachment in Ureteral Stents

**DOI:** 10.3390/mi11040408

**Published:** 2020-04-13

**Authors:** Antonio De Grazia, Gareth LuTheryn, Alireza Meghdadi, Ali Mosayyebi, Erika J. Espinosa-Ortiz, Robin Gerlach, Dario Carugo

**Affiliations:** 1Bioengineering Science Research Group, Faculty of Engineering and Physical Sciences, University of Southampton, Southampton SO17 1BJ, UK; A.DeGrazia@soton.ac.uk (A.D.G.); G.LuTheryn@soton.ac.uk (G.L.); A.Meghdadi@soton.ac.uk (A.M.); A.Mosayyebi@soton.ac.uk (A.M.); 2Institute for Life Sciences (IfLS), University of Southampton, Southampton SO17 1BJ, UK; 3Center for Biofilm Engineering and Department of Chemical and Biological Engineering, Montana State University, Bozeman, MT 59717, USA; erika.espinosaortiz@montana.edu (E.J.E.-O.); robin_g@montana.edu (R.G.)

**Keywords:** ureteral obstruction, ureteral stent, microfluidics, stent-on-a-chip, bacterial attachment, biofilm formation, cavity flow, wall shear stress, CFD simulations

## Abstract

Obstructions of the ureter lumen can originate from intrinsic or extrinsic factors, such as kidney stones, tumours, or strictures. These can affect the physiological flow of urine from the kidneys to the bladder, potentially causing infection, pain, and kidney failure. To overcome these complications, ureteral stents are often deployed clinically in order to temporarily re-establish urinary flow. Despite their clinical benefits, stents are prone to encrustation and biofilm formation that lead to reduced quality of life for patients; however, the mechanisms underlying the formation of crystalline biofilms in stents are not yet fully understood. In this study, we developed microfluidic-based devices replicating the urodynamic field within different configurations of an occluded and stented ureter. We employed computational fluid dynamic simulations to characterise the flow dynamic field within these models and investigated bacterial attachment (*Pseudomonas fluorescens*) by means of crystal violet staining and fluorescence microscopy. We identified the presence of hydrodynamic cavities in the vicinity of a ureteric occlusion, which were characterised by low levels of wall shear stress (WSS < 40 mPa), and observed that initiation of bacterial attachment occurred in these specific regions of the stented ureter. Notably, the bacterial coverage area was directly proportional to the number of cavities present in the model. Fluorescence microscopy confirmed that the number density of bacteria was greater within cavities (3 bacteria·mm^−2^) when compared to side-holes of the stent (1 bacterium·mm^−2^) or its luminal surface (0.12 bacteria·mm^−2^). These findings informed the design of a novel technological solution against bacterial attachment, which reduces the extent of cavity flow and increases wall shear stress over the stent’s surface.

## 1. Introduction

Ureters are conduits that connect kidneys to the bladder, allowing for the urine produced by the kidneys to drain through the distal urinary system. Under physiological conditions, peristaltic contractions of the ureteral wall govern transport of urine towards the bladder. However, in some pathological cases, localised shrinkage of the cross-sectional area of the ureter can occur, impairing the physiological passage of urine. This can be due to a number of conditions, such as kidney stones lodging within the ureteral lumen, tumours compressing the ureteral wall, or inflammatory strictures (Figure 1a) [1,2,3]. If not treated, a ureteral obstruction can lead to several health complications. In order to prevent or alleviate these complications, clinicians often recur to the deployment of a ureteral stent. Stents are flexible hollow tubes that provide an alternative pathway for the urine to bypass ureteral obstructions. A common stent design includes two double-J ends to prevent stent migration, and side-holes punched through its wall in order to promote urinary drainage and bypass the obstructed ureteral section [4,5,6,7]. Although they are widely used in clinical settings, normal ureteral stent function can be compromised by problems or side effects, including urothelial irritation, stent migration, encrustation due to attachment of crystalline particles, and biofilm formation due to bacterial colonisation [8,9,10]. In addition to reducing a patient’s quality of life, these complications inevitably lead to an increased economic burden, since stent replacement, additional clinical procedures, and aftercare may be required [11,12].

Bacteria are able to adhere to both biotic materials, such as human tissues, and abiotic surfaces, including metals, plastics, and other materials that are commonly used in medical devices. The initial interaction and attachment of bacteria on these surfaces can lead to biofilm formation, which is a complex three-dimensional (3D) structure of bacteria encased in a self-produced extracellular matrix. The latter typically comprises of lipids, polysaccharides, nucleic acids, and proteins, which provide mechanical stability and attachment strength to biofilms while protecting bacteria from external physical and chemical threats [13]. Planktonic and sessile bacteria are already well equipped to develop genetic resistance to antimicrobial agents. However, a significant complication of biofilm development is that the extracellular matrix acts as a further physical barrier to antimicrobial agents and subsequently confers an enhanced tolerance to treatment [14]. Moreover, other characteristics of biofilms that provide them with enhanced survival and resistance to antimicrobials include social cooperation and localised gradients within the biofilm that provide habitat diversity [15]. Bacterial attachment leading to biofilm formation is a critical step in biofilm development and organisation. In this context, a better understanding of the influence of wall shear stress on bacterial attachment is fundamental to prevent issues related to the growth of damaging biofilms over ureteral stents.

Wall shear stress (WSS) describes the tangential force per unit area applied over a solid boundary by a fluid in motion. It is well established that hydrodynamic forces, such as WSS, play a crucial role in bacterial attachment, including the carriage of planktonic bacteria towards a surface and their subsequent interaction. Attachment mechanics can be influenced by fluid dynamics with uncertain effects. Notably, it has been reported that low wall shear stress levels (<90 mPa) may promote bacterial attachment, whereas increased wall shear stress under turbulent flow (3.7–7.3 Pa) may prevent long-lasting attachment [16]. In another study, it has been shown that increasing WSS resulted in less frequent attachment events, but the number of firmly attached bacteria increased. The highest level of attachment was observed at WSS of 50 mPa, in a tested range of 50 mPa to 10 Pa [17]. Nevertheless, although there are studies reporting on the effect of WSS on bacterial attachment and subsequent biofilm formation, the magnitude of WSS is possibly not the only determinant of attachment.

Microfluidic technology has demonstrated significant potential as a versatile tool for investigation of the mechanisms of biofilm formation upon exposure to different flow dynamic conditions, allowing testing of different channel architectures and types of materials or surfaces [18]. In a previous study, Mosayyebi et al. [19] developed a stent-on-a-chip device to investigate deposition of crystals in a ureteral stent. Using this model, regions of stents that are prone to particle deposition could be identified, which was attributed to stagnant flow and corresponding low WSS levels. Regions subject to particle deposition include the hydrodynamic cavity that is formed in proximity to a ureteral obstruction, and stent side-holes that do not actively contribute to urinary drainage (referred to as “inactive” side-holes). In a more recent study, Mosayyebi et al. [20] proposed that changing the side-hole architecture into a streamlined shape significantly reduces the amount of particle deposition when compared to conventional side-holes. These studies improved our understanding of the role of flow dynamics on the deposition and attachment of encrusting particles in ureteral stents.

Building upon these earlier investigations, in the present work we employed a microfluidic-based model of a stented and occluded ureter to evaluate short-term attachment of bacterial cells in ureteral stents. Combining computational fluid dynamic (CFD) simulations and fluorescence microscopy, we identified hydrodynamic regions of a stent that are more susceptible to bacterial attachment. Results from this study may inform the design of novel stent architectures, resulting in increased lifetime and greater resistance against biofilm formation.

## 2. Materials and Methods

### 2.1. Stent-on-a-Chip Devices: Design and Manufacturing

Microfluidic devices (referred to as stent-on-a-chip, SoC) were designed to replicate key flow dynamic features of a stented ureter in the presence of different types of ureteral obstruction (Figure 1a). The rationale behind the SoC design is described in greater detail by Mosayyebi et al. [20] and is illustrated in Figure 1a. Briefly, the compartment of the stent was modelled by a straight 1.5-mm wide channel, with a wall thickness of 1 mm. Dimensions were taken from a commercially available double-J stent (model Universa^®^, Cook^®^ Medical, Bloomington, IN, USA). The wall of the stent was designed to contain two side-holes (0.8 mm wide), positioned 19 mm away from each other. The side-hole closer to the device’s inlet is herein referred to as “proximal”, whilst the other side-hole is referred to as “distal”. The ureter was modelled as a 5-mm wide channel [21]. The height of all channels was set to 0.75 mm. The overall longitudinal distance between the inlet and outlet of the device was of 35 mm, which was chosen to ensure that the design could fit on a glass slide/coverslip for high-resolution microscopy imaging.

A replica moulding technique was employed in order to manufacture SoC devices. Positive moulds of the model architecture were designed in SolidWorks CAD software, as shown in Figure 1b (CAD design), and subsequently 3D printed in polylactic acid (PLA) using an Ultimaker 2+ printer, as shown in Figure 1b (3D printed moulds). Soft lithography was then employed to generate a channel architecture for the SoC geometry within a layer of polydimethylsiloxane (PDMS). In particular, PDMS was initially formed by mixing silicone elastomer and curing agent (SYLGARD™ 184, Dow Corning Corporation, Midland, MI, USA) in a 10:1 proportion by weight. Liquid PDMS was then poured over the positive moulds, degassed, and left to cure at room temperature overnight to avoid undesired deformation of the PLA at higher temperatures. Afterwards, the PDMS layer was bonded to a glass slide (length: 75 mm, width: 25 mm, thickness: 1 mm, Corning^®^) or to a coverslip for high-resolution microscopy imaging (length: 60 mm, width: 24 mm, thickness: 0.13–0.16 mm, Thermo Scientific Menzel^TM^, Saarbruckener, Germany). Bonding was performed via corona treatment, using a handheld corona machine (BD-20AC Laboratory Corona Treater, Electro-Technic Products, Chicago, IL, USA).

Figure 1b (CAD designs) shows the four different SoC geometries employed in this study. In particular, the unobstructed design replicates the stented ureter in the absence of obstruction; the obstructed I design includes an obstruction of the lumen of the ureter located 4.5 mm away from the proximal side-hole (i.e., this results in one cavity being formed); the obstructed II design includes an obstruction of the lumen of the ureter located in between the two side-holes (i.e., this results in two cavities being formed); and the obstructed III design models a scenario where the occlusion is located in very close proximity to the proximal side-hole (i.e., no cavity is formed in this case).

### 2.2. CFD Simulations of the Flow Field within SoC Models

CFD simulations were conducted using ANSYS Workbench software package version 19.1 (academic license). Geometries were meshed using ANSYS Meshing, CFD simulations were carried out using FLUENT, and postprocessing was done using CFD-Post. Simulations were used to determine the spatial distribution of WSS over the bottom surface of the device, which corresponds to the microscope’s focal plane in subsequent experiments.

The appropriate mesh size for the geometries in Figure 1 (and later in Figure 5) was selected after performing a mesh sensitivity study. ANSYS DesignXplorer was utilised to define the different mesh sizes evaluated in the mesh sensitivity study. Meshes were created using a patch independent tetrahedron method to optimise skewness and orthogonal quality of elements. This approach allows faster computation time and quick solution convergence. Global element sizes were set to 400, 300, 200, 160, 120, 100, 90, and 80 µm. Additionally, proximity and curvature refinements were enabled with a minimum size limit of 40 µm to allow for size refinement in corners and edges of geometries. Assessment of the mesh sensitivity study was performed with respect to computation time and convergence of velocity profiles along certain defined linear locations within the narrowest channels of the geometries, yielding an optimum element size of 100 µm, with an average skewness of 13.7% and average orthogonal quality of 86.2% (see Figure A1). This element size was assumed to be the optimum mesh size for all the geometries in this study, given that the width of the narrowest channels in all designs was identical.

Identical boundary conditions were defined for each geometry. The continuous phase material (i.e., fluid) was set to water (density: 997.044 kg⋅m^−3^, dynamic viscosity: 0.001 Pa⋅s). The volumetric inlet flow rate was set to 1 mL⋅min^−1^, while a 0 gauge pressure boundary condition was imposed at the device outlet. The imposed value of volumetric flow rate was chosen to replicate a physiologically relevant condition, in accordance with previous studies [20,21,22,23]. A no-slip boundary condition was set at the channel walls. ANSYS FLUENT was employed to solve mass and momentum conservation equations over the fluidic domain, using the IRIDIS 4 supercomputer cluster at the University of Southampton. Every simulation was run for 3000 iterations, at the end of which solution residuals were below 1 × 10^−5^.

### 2.3. Preparation of Liquid Bacterial Culture

*Pseudomonas fluorescens* was used as a bacterial model in this study. Although this bacterium is not a common pathogen found in the urinary tract, it was chosen as a model for a proof-of-concept investigation of bacterial attachment. A sterile inoculating loop was used to select three identical colonies of *Pseudomonas fluorescens (ATCC 13525)*, grown on *Pseudomonas* isolation agar (17208, Sigma-Aldrich, Gillingham, UK). The colonies were inoculated into 5 mL of tryptic soy broth (TSB) (22092, Sigma-Aldrich) and incubated at 37 °C overnight. The turbid overnight suspension was diluted 1:100 (approximately 10^6^ bacteria/mL) in sterile TSB, and this suspension was used to inoculate the SoC.

### 2.4. Assessment of Bacterial Accumulation within SoCs by Crystal Violet Biomass Staining and Live/Dead Fluorescent Labelling

SoCs were flushed with 10 mL of 70% ethanol, and Milli-Q water was then administered at a volumetric flow rate of 1 mL⋅min^−1^ for 10 min using a syringe pump (PHD 2000, Harvard Apparatus, Cambourne, UK). A 60 mL syringe (BD Plastipak Syringe, Thermo Fisher Scientific, Waltham, MA, USA) was used to draw up the diluted suspension for administration into the sterilised SoC. A suspension of *Pseudomonas fluorescens* was diluted 1:100 in TSB and was allowed to flow continuously for 60 min at 1 mL⋅min^−1^. After 60 min, the flow of the bacterial suspension was halted for 15 min, during which 1% (*v/v*) crystal violet (Pro-Lab Diagnostics™ PL8000, Richmond Hill, Ontario, Canada) was pipetted into each SoC to fully prime the channels. Afterwards, Milli-Q water was used to flush the crystal violet solution away from the channels at 1 mL⋅min^−1^, until the fluid exiting the device was visibly clear. For control tests, sterile TSB was injected through the SoCs, and the devices were also stained with crystal violet in the same manner, but without the presence of bacteria to look at non-specific staining of PDMS. Photographs of the crystal-violet-stained SoC channels were taken using a digital single-lens reflex camera (Canon EOS 6D, 20.2 megapixels). Images were analysed in ImageJ [24]; each image was binarised so that the detectable violet colour would turn into black and the background would become white. The total areas occupied by both black and white pixels were automatically quantified, and data were presented as the average value and corresponding standard deviation. Experiments for the control and the four SoC designs were repeated three independent times. Because of the different priming volumes in the four designs, the bacterial attachment area (B_A_) was expressed in percentage as follows:(1)$$B_{A}\;\left(\%\right)=\frac{A_{C}}{A_{T}}\;\cdot \;100$$
where *A_C_* is the area stained by crystal violet and *A_T_* is the total considered area of the SoC geometry. Values of *A_T_* for the different SoC designs are reported in Table 1.

The green and red fluorescent nucleic acid stains SYTO™ 9 (Invitrogen™) and propidium iodide (PI) (Invitrogen™, P3566), respectively, were used to assess the viability of a planktonic suspension of *P. fluorescens* within the SoC. SYTO™ has a fluorescence excitation maximum at 483 nm and an emission maximum at 503 nm and binds to DNA in live cells, whereas PI has an excitation maximum at 535 nm and an emission maximum at 617 nm and binds to DNA of dead cells.

SYTO™ 9 and PI were administered into the SoC in a 1:1 volume ratio, with final concentrations of 2.5 µM and 15 µM, respectively. The SoC was protected from the light, while the fluorescent stains were permitted to interact with the bacterial suspension for a minimum of 3 min; the channels were subsequently flushed of excess stain and the chip was observed by fluorescent microscopy (Invitrogen™ EVOS M5000, Thermo Fisher Scientific, Loughborough, UK). Overview microscopy images were taken with a 20× objective and were stitched together, then a 100× oil immersion objective was used to take high-resolution images in regions of interest. All high-resolution images were taken at the same focal plane (bottom surface of the SoC).

### 2.5. Statistical Analysis

The experimental data for bacterial attachment are presented as average ± standard deviation. Differences between the unobstructed design and the three obstructed designs were assessed using unpaired Student’s *t*-test. The significance level was set to 0.05, and differences were considered statistically significant for *p* < 0.05.

## 3. Results

### 3.1. Ureteral Obstructions Promote Bacterial Attachment

Figure 2 shows the WSS distribution over the bottom surface of the unobstructed and obstructed I designs, as determined from CFD simulations, along with corresponding images of crystal violet staining, highlighting regions of greater bacterial attachment.

Results from the unobstructed device in the absence of a ureteral occlusion showed no evidence of short-term bacterial attachment in any region of the model, including within regions of low wall shear stress (WSS < 40 mPa), such as side-holes (Figure 2a). All positive controls showed absence of non-specific staining on the SoC inner surfaces (see Figure A2 for a representative control test, using the unobstructed SoC model).

As for the obstructed I SoC device (i.e., with an occlusion located 4.5 mm away from the proximal side-hole), CFD simulations showed very low levels of wall shear stress within the hydrodynamic cavity formed in proximity to the occlusion (WSS << 40 mPa). This region is also characterised by the presence of laminar eddies, as previously reported [20,22]. Moreover, simulations revealed that the presence of a ureteral occlusion diverted a greater amount of fluid flow through the proximal side-hole, resulting in increased WSS levels in this hole compared to the unobstructed case. Upon staining bacteria with crystal violet, it was observed that the cavity region and the nearby proximal side-hole exhibited greater levels of bacterial attachment compared to other regions of the model. Crystal violet staining for the whole SoCs (unobstructed and obstructed I) are shown in Figure A2.

To further investigate the role of cavity flow on the initiation of bacterial attachment, two additional SoC designs were tested. These are labelled as “obstructed II with two cavities”, in which the obstruction of the ureter was located between the two neighbouring side-holes; and “obstructed III with no cavities”, in which the obstruction was positioned at the edge of the proximal side-hole, hence resulting in the absence of cavity flow.

As shown in Figure 2, numerical simulations showed that cavities formed by a ureteral occlusion were characterised by the lowest level of wall shear stress (WSS << 40 mPa), which is consistent with the obstructed I design. When the occlusion was located between neighbouring side-holes (as in the “obstructed II with double cavity” design), two cavities were formed (i.e., distally and proximally to the occlusion; Figure 3a). Both cavities were characterised by bacterial attachment, as revealed by crystal violet staining, and attachment also occurredwithin the nearby side-holes (Figure 3a). Removing the presence of cavity flow by positioning the obstruction at the edge of the proximal side-hole resulted in the absence of detectable bacterial attachment within the proximal side-hole and surrounding regions of the model (as observed in the obstructed III design).

The average bacterial coverage area (determined from crystal violet staining) was equal to 17.5% ± 2.6% for the obstructed SoC (obstructed I), whilst no measurable coverage area was determined for the unobstructed SoC (Figure 2b). On the other hand, the average coverage area was equal to 38.1% ± 3.2% for the obstructed II design with double cavity (Figure 3b), which is 2.2 times greater than in the obstructed II design (with single cavity). This may suggest a direct proportionality between bacterial coverage area (as quantified from crystal violet staining) and the number of cavities. Statistical analysis showed a significant difference between bacterial attachment in the unobstructed design (null) compared to the obstructed I and obstructed II designs, whereas there was no statistical difference between the unobstructed and obstructed III designs.

### 3.2. Fluorescence Microscopy Shows Greater Attachment of Bacteria Close to a Ureteric Occlusion

Crystal violet staining showed significant bacterial attachment within the hydrodynamic cavities located nearby ureteral occlusions, which also extended to influence neighbouring regions (including side-holes of the stent). In order to obtain additional insights about the spatial distribution of bacteria in these regions, fluorescence microscopy images of fluorescently labelled bacteria in the obstructed I SoC device were acquired (Figure 4). The obstructed I device was chosen as a representative model for imaging of bacterial attachment in regions where crystal violet staining was observed. After 1 h, bacteria maintained detectable SYTO™ 9 staining whilst propidium iodide staining was absent. Moreover, images revealed that there was a difference in the bacterial number density between the cavity region, the side-hole, and the segment of stent lumen closer to the SoC inlet. Bacteria were present at greater number density (3 bacteria·mm^−2^) in the cavity, which is the region characterised by the lowest wall shear stress levels (WSS << 40 mPa) and the presence of laminar vortices (Figure 4a). The bacterial density decreased to 1 bacterium·mm^−2^ (Figure 4b) in the side-hole (WSS ~85 mPa), and to 0.12 bacteria·mm^−2^ (Figure 4c) in a region of the stent lumen located in proximity to the SoC inlet (WSS = ~130 mPa). Generally, the bacterial number density was greater in proximity to the side walls of the device.

## 4. Discussion

Ureteral stents are deployed clinically to restore urinary drainage from kidneys to the bladder when drainage is impaired by pathological ureteral occlusions; however, they can suffer from complications, leading to side effects. One of the most recurrent complications is the formation of mineral precipitating biofilms on the surfaces of the stent, possibly causing failure of the stent and urinary tract infection, which might require pharmacological treatment or surgical intervention [25].

In the present study, we employed microfluidic-based models (referred to as stent-on-a-chip, SoC) to investigate the initiation of bacterial attachment in the stented and occluded ureter. Our findings may suggest that one of the factors governing bacterial attachment in the stented ureter is the presence of cavity flow in areas located in the proximity of a ureteral obstruction. Cavity flow originates in the region located between a complete occlusion of the ureter lumen and one or more stent side-holes, and is characterised by low WSS levels (WSS << 40 mPa) and the formation of low-velocity laminar vortices. The combination of these conditions may promote attachment of bacterial cells present in urine.

This hypothesis was tested through evaluation of four different SoC designs. One design modelled the scenario of an unobstructed and stented ureter (unobstructed design, see Figure 2); in this geometry, cavity flow is absent and both side-holes are inactive (i.e., no net fluid drainage occurs through these holes). Staining with crystal violet did not show any detectable bacterial attachment in this device, including within inactive side-holes, where the average WSS is << 40 mPa. These findings appear to differ from the recent studies by Mosayyebi et al. [20], which showed deposition of encrusting particles within inactive side-holes in a similar stent-on-a-chip device. It should, however, be noted that the amount of particle deposition and attachment over a surface (within a given fluidic domain) likely depends on both mass flow rate of the particle (i.e., mass of flowing particles per unit time), sedimentation rate, and the level of WSS over the surface. Thus, it is strongly dependent on the particles’ velocity, size, volumetric density, and concentration. Bacterial cells are smaller than the encrusting particles observed by Mosayyebi et al. [19], and also have a significantly lower density (~1.1 g cm^−3^) compared to crystals commonly present in urine (~2.2 g cm^−3^ for calcium oxalate). Interestingly, comparing findings from these studies suggest that the temporal and spatial dynamics of bacterial and crystal attachment within ureteric stents may be different, given the significant differences in the physical properties and flow behaviour of these particles. Taken together, these studies suggest that different regions of a stent may be subject to the formation of different types of encrustation and crystalline biofilms (i.e., with different architecture and composition), depending on the local flow dynamic conditions.

Absence of bacterial attachment was also observed in the stent-on-a-chip device with the obstruction positioned in very close proximity to the proximal side-hole (obstructed III). This device configuration was characterised by the absence of cavity flow, and experiments shown in Figure 3 (obstructed III case) may suggest that WSS levels were high enough (WSS > 40 mPa) to prevent initiation of bacterial attachment.

Designs that exhibited cavity flow, either in the form of a single cavity (obstructed I, with the occlusion located 4.5 mm before the proximal side-hole) or two cavities (Obstructed III, with the occlusion located between two neighbouring side-holes), showed enhanced bacterial attachment within the cavity region. Interestingly, the attachment area appeared to be directly proportional to the number of cavities. These observations are consistent with findings from a previous fluid dynamic investigation by Clavica et al. [22] on a three-dimensional artificial model of the stented ureter, and the more recent studies by Mosayyebi et al. [20] that showed significant levels of crystal deposition within cavities. In this study, bacteria were also detected in regions located near the cavity, including the active side-hole and the stent’s lumen, although they were present at a lower density compared to the cavity itself (as revealed by fluorescence microscopy, Figure 4). Although the underlying temporal dynamics of the process is not yet fully understood, it could be hypothesised that bacteria initially deposited and attached onto the bottom surface of the cavity, given the presence of low-velocity vortex flow in this region, and subsequently migrated towards other neighbouring regions, including side-holes of the stent. Notably, upstream bacterial colonisation within an in vitro model of the urinary system was recently observed by Hobbs et al. [23], demonstrating bacteria’s ability to migrate against the flow at comparable levels of volumetric flow rate.

The findings discussed in this paper are also in qualitative agreement with clinical studies, demonstrating enhanced presence of biofilms on the outside surface of stents [25], which could be potentially attributed to the presence of obstructions (or strictures) between the ureter and stent.

Based upon the results from this study, a potential design solution is herein proposed that could reduce the amount of bacterial attachment within the stented ureter. In order to reduce the extent of cavity flow, several side-holes could be manufactured near a ureteral occlusion (both proximally and distally), where the location of occlusion could be determined on a patient-specific basis from medical images (i.e., MRI scans). These multi side-hole designs have been evaluated through CFD simulations (Figure 5), by applying this design approach to the SoC models shown earlier in Figure 2 (obstructed I) and Figure 3 (obstructed II), demonstrating potential to enhance drainage and increase WSS in the vicinity of a ureteral occlusion. By adding multiple side-holes in proximity to the occlusion, WSS at the cavity increased from <<40 mPa (as seen in Figure 2 and Figure 3) to values > 40 mPa, which is likely to prevent short-term bacterial attachment.

## 5. Limitations and Future Perspectives

The experimental setup used in this study presents some limitations: (1) The microfluidic devices are only simplified replicas of the human stented ureter. However, they represent a high-throughput engineering approach to the identification of fluid dynamic regions of a stent that may be subject to bacterial attachment. (2) The vertical thickness of the microfluidic channels (750 µm) prevented high-resolution and time lapse imaging of bacterial attachment. Fabricating thinner channels (1–5 µm) might provide more information about the bacterial distribution in the volume of the device rather than just a fixed focal plane. (3) Bacteria used in this study (*P. fluorescens*) are not characteristic of the biofilm population in urinary tracts. Further tests with *P. aeruginosa* or *Proteus mirabilis* and artificial urine would provide further understanding of the relationship between bacterial attachment and biofilm formation, as well as crystal formation and encrustation. (4) CFD simulations provided insights into the relationship between flow metrics (such as WSS) in a stented and occluded ureter and short-term bacterial attachment. Further investigations will focus on identifying the individual contribution of different factors governing bacterial deposition and attachment to the inner surfaces of the model. Similarly, experiments should be conducted to investigate the effect of varying the length of the cavity and other geometrical properties of the stent. (5) This study focuses on short-term bacterial attachment in order to identify regions of a stent where biofilm formation may be initiated. Future investigations could focus on longer timescales.

## Figures and Tables

**Figure 1 micromachines-11-00408-f001:**
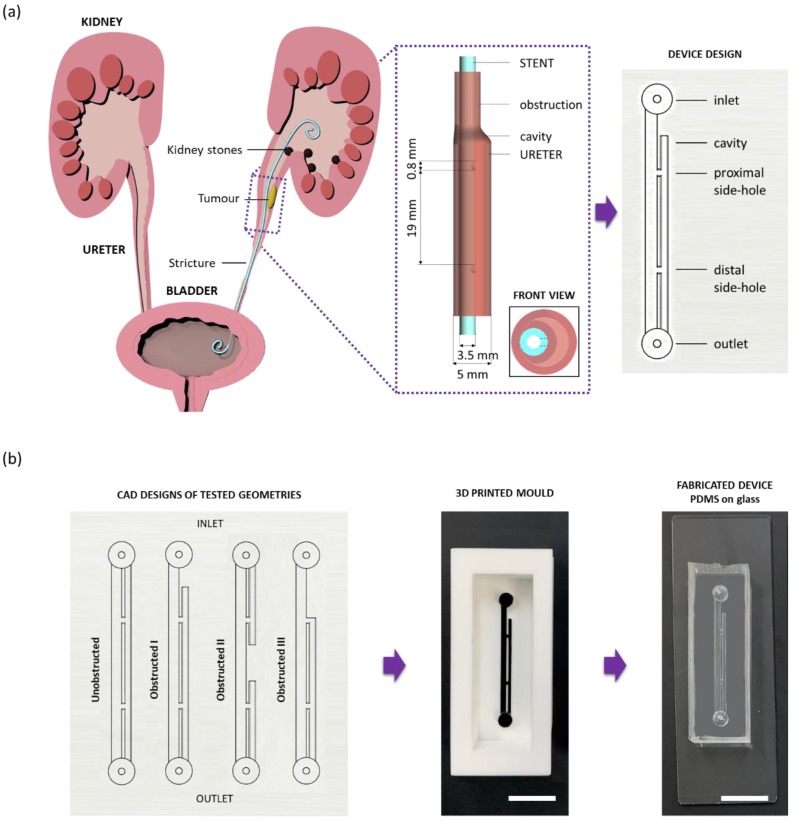
Illustration of the design rationale and manufacturing steps for the stent-on-a-chip (SoC) devices. Ureteral stents are clinically deployed to promote drainage of urine from kidneys to the bladder in cases where the ureter passage is compromised. (**a**) Occlusions such as tumours, kidney stones, and strictures can block the ureter lumen, meaning that urine cannot naturally flow from kidneys to the bladder via peristalsis. A ureteral stent is a hollow tube with several side-holes positioned along its length that are designed to promote urinary flow around the occlusion. In the framed zoomed-in section, key dimensional properties of a typical double-J stent are reported. The SoC device provides a replica of a region of the stented ureter located in proximity to the occlusion. (**b**) SoC devices were designed so as to model either an unobstructed stented ureter or an obstructed stented ureter, with the obstruction located at different positions within the ureter. Positive moulds of the SoC design were then 3D printed and employed to manufacture polydimethylsiloxane (PDMS) layers containing the SoC architecture. These layers were bonded to glass slides or coverslips in order to obtain a sealed SoC device that could be employed to monitor bacterial attachment in ureteral stents via integration with optical microscopy. Scale bars: 3 mm.

**Figure 2 micromachines-11-00408-f002:**
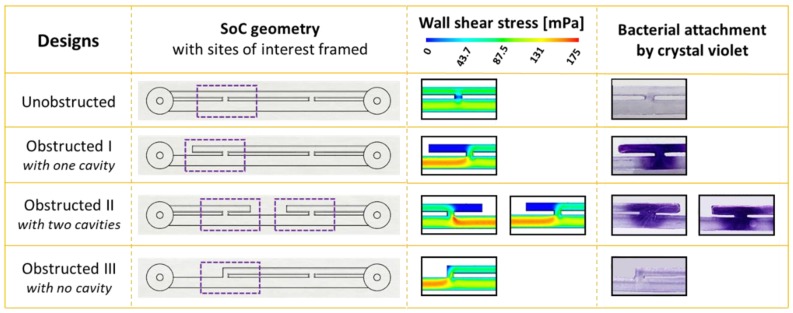
Bacterial attachment occurs in proximity to the hydrodynamic cavity formed by a ureteral occlusion. The unobstructed SoC device does not show evidence of bacterial attachment, as indicated by the absence of detectable crystal violet staining. The obstructed I SoC device (i.e., with one cavity) shows bacterial attachment in the cavity region, which corresponds to an area of low wall shear stress (WSS) and laminar vortices. The spatial distribution of WSS over the bottom surface of the devices (determined from computational fluid dynamic (CFD) simulations) and images of crystal violet staining are illustrated for those regions affected by bacterial attachment (purple frames). Average WSS values are generally relatively high (WSS ~ 133 mPa) inside the stent lumen and become significantly lower (WSS < 40 mPa) within side-holes in the unobstructed design and within the occluded cavity in the obstructed I design. However, crystal violet staining was only visible within the cavity. The SoC device named “obstructed II with two cavities” showed attachment of bacteria in both cavities, located both proximally and distally to the occlusion. The SoC device named “obstructed III design with no cavities” (i.e., with the obstruction located at the edge of the proximal side-hole) presented no visible attachment, suggesting that wall shear stress was not low enough to initiate bacterial attachment. CFD simulations showed low wall shear stress (WSS << 40 mPa) in both cavity regions for the obstructed II design, whereas it was always greater than 40 mPa in the obstructed III design.

**Figure 3 micromachines-11-00408-f003:**
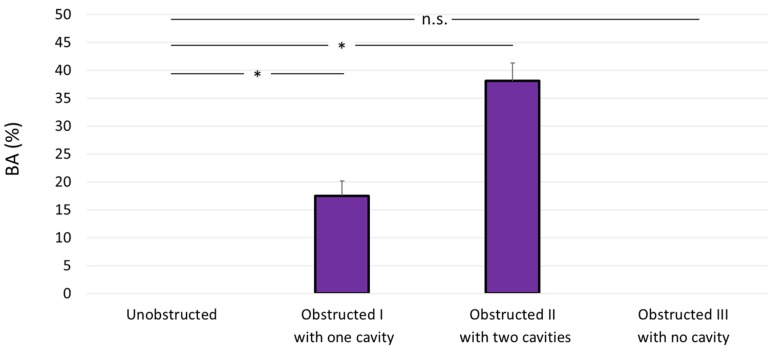
Bacterial attachment directly correlates with the number of cavities. The bacterial coverage area (BA) was calculated by analysis of crystal violet images, considering the entire channel volume of the chip for all designs. The unobstructed and “obstructed III with no cavity” designs show null coverage area, whereas the obstructed I and obstructed II designs show an average value of 17.5% ± 2.6% and 38.1% ± 3.2%, respectively. Data are reported as average ± standard deviation (*n* = 3). Note: * *p* < 0.05.

**Figure 4 micromachines-11-00408-f004:**
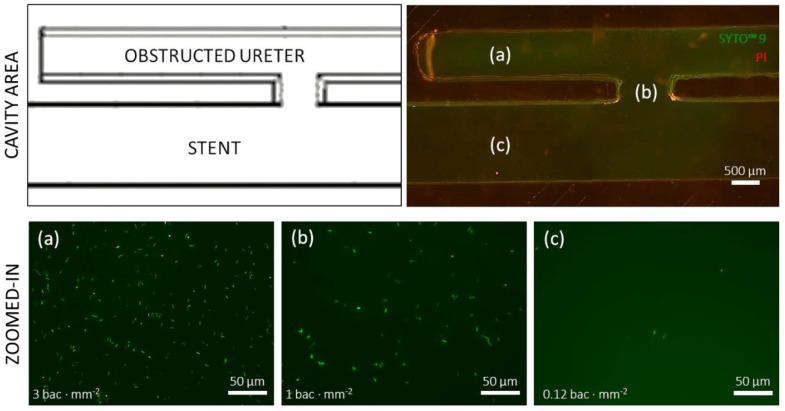
Fluorescence staining by SYTO™ 9 and propidium iodide reveals differences in the number density of bacteria adhered over different regions of interest in the obstructed I device. (Top) Close up schematic of cavity area and overview fluorescence image of SYTO™ 9 (green) and propidium iodide (PI; red). (Bottom) Fluorescence micrographs were obtained to quantify the number density of bacteria in (**a**) the cavity region (average of 238 ± 7 bacteria in the field of view), (**b**) the proximal side-hole (average of 80 ± 10 bacteria in the field of view), and (**c**) a region of the stent lumen close to the inlet (average of 10 ± 4 bacteria in the field of view). All images were acquired in the same focal plane (top surface of the glass coverslip).

**Figure 5 micromachines-11-00408-f005:**
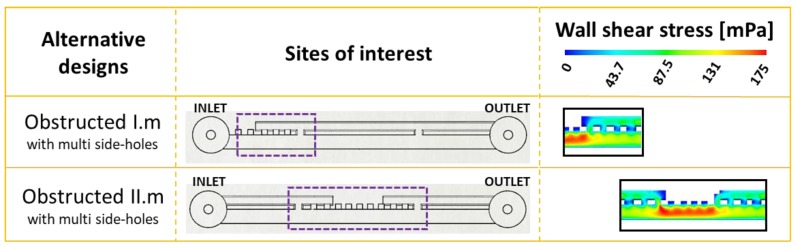
Manufacturing multiple side-holes in proximity to a ureteral obstruction decreases the possibility of bacterial attachment by enhancing fluid drainage and wall shear stress levels near the occlusion. The same designs that caused bacterial attachment (obstructed I and obstructed II) were modified to contain multiple side-holes close to the obstruction. As assessed by CFD simulations, the addition of side-holes increases the magnitude of WSS in regions that were previously subject to cavity flow.

**Table 1 micromachines-11-00408-t001:** Total area of the four tested SoC designs.

SoC Design	A_T_ (mm^2^)
Unobstructed	92.5
Obstructed I	88.0
Obstructed II	87.5
Obstructed III	83.8

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
