# Peer review of "A Microfluidic-Based Investigation of Bacterial Attachment in Ureteral Stents"

_micromachines, 2020, doi:10.3390/mi11040408_

Round 1

Reviewer 1 Report

An interesting manuscript investigating the relationship of fluid dynamics and wall shear stress to attachment of bacterial cells to ureteral stents within microfluidic-based models of obstructed and non-obstructed ureters. The influence of the occlusion site on resultant adherence was, in addition, investigated through the use of stents-on-a-chip with various configurations.

The manuscript has been very well written and addresses an important clinical problem of stent blockages, with the findings informing a novel stent design with promising potential to address this issue.

Some minor comments to be considered:

Line 90 – include author name and place [20] after name.

Line 91 – describe the findings in past tense.

Line 110 – was the ureter modelled as a 3 mm wide channel?  From Figure 1 the width appears to be 5 mm (but perhaps I have misunderstood!).

Line 128 – change ‘like’ to ‘such as’.

Line 310 – caption of Figure 4 needs (b) and (c) inserted.

Line 367 – change ‘presence obstructions’ to ‘presence of obstructions’

Figure A2 – it would be useful to also include images of the control obstructed SoCs to ensure that the observed CV staining is not a result of poor flushing in regions of low WSS.  Alternatively, in Figure 3, the area stained by CV in the presence of bacteria relative to the area stained by CV in the controls could also be presented for each design.  

Author Response

R: Line 90 – include author name and place [20] after name.

A: This has been edited accordingly.

R: Line 91 – describe the findings in past tense.

A: We have changed the verb’s tense accordingly.

R: Line 110 – was the ureter modelled as a 3 mm wide channel?  From Figure 1 the width appears to be 5 mm (but perhaps I have misunderstood!).

A: Thank you for pointing this out. This was a typo and has now been corrected.

R: Line 128 – change ‘like’ to ‘such as’.

A: This has been changed accordingly.

R: Line 310 – caption of Figure 4 needs (b) and (c) inserted.

A: Thank you; captions have now been inserted.

R: Line 367 – change ‘presence obstructions’ to ‘presence of obstructions’

A: This has now been corrected.

R: Figure A2 – it would be useful to also include images of the control obstructed SoCs to ensure that the observed CV staining is not a result of poor flushing in regions of low WSS.  Alternatively, in Figure 3, the area stained by CV in the presence of bacteria relative to the area stained by CV in the controls could also be presented for each design.  

A: We thank the reviewer for their comment. We can confirm that no staining could be visually detected in all control tests performed, for the different types of model investigated. This was consistent with the control test image reported in the Appendix. We have not taken photographs of all these control tests, but have now added a comment on this point in the revised manuscript.

Reviewer 2 Report

The authors have shown a microfluidic model of uretteral obstruction with numerical simulation and experimental results. 

The technical writing is fluent and sound. 

However, some technical parts may require correction.

  1. page 3 line 120, is the PDMS indeed is mixed at 1:10 monomer and curing agent but not 10:1?
  2. page 3 line 124, it is believed that oxygen plasma is restricted to charged gas molecule at low vacuum. I think a corona treatment will suffice.
  3. page 6 DLSR camera uses CMOS sensor which should not be mistaken with CCDs. 
  4. It was shown that PDMS devices were bonded to 1mm thick glass but on page 6 a 100x oil immersion objective was used but to my knowledge that the working distance will not be sufficient to be seen with a 100x objective. Please double check technical information. 

The authors have discussed extensively the limitations of the study and the future perspectives. The hydrodynamic stagnant regions of the ureteral stent can retain bacteria and allow adhesion to form as experimentally shown in this study. It is also interesting to show that in case of obstructed devices, at the high shear region near the cavity region, a large amount of bacteria adhesion is still observed.

A more clinical relevant bacteria strain, more biomimetic surfaces, and longer experiment to allow biofilm formation can be more attractive to readers. 

Author Response

R: page 3 line 120, is the PDMS indeed is mixed at 1:10 monomer and curing agent but not 10:1?

A: Thank you for pointing this out; the ratio was indeed 10:1 (monomer to curing agent), and the text has now been corrected.

R: page 3 line 124, it is believed that oxygen plasma is restricted to charged gas molecule at low vacuum. I think a corona treatment will suffice.

A: We have edited the text following the reviewer’s suggestion.

R: page 6 DLSR camera uses CMOS sensor which should not be mistaken with CCDs.

A: Thank you for pointing this out. We have now edited the text accordingly.

R: It was shown that PDMS devices were bonded to 1mm thick glass but on page 6 a 100x oil immersion objective was used but to my knowledge that the working distance will not be sufficient to be seen with a 100x objective. Please double check technical information.

A: Apologies if this has not been clearly described in the original submission. We have used 1 mm thick glass slides for crystal violet experiments, and 0.13-0.16 mm thick coverslips for high-resolution fluorescence microscopy. The manuscript has now been edited to clarify this.